# Constructing Precipitable Water Vapor Map from Regional GNSS Network Observations without Collocated Meteorological Data for Weather Forecasting

Biyan Chen[1,2,3], Wujiao Dai[1,2,3], Zhizhao Liu[4], Lixin Wu[1,3], Cuilin Kuang[1,2,3], Minsi Ao[5]

[1]School of Geosciences and Info-Physics, Central South University, Changsha, Hunan, China
[2]Key Laboratory of Precise Engineering Surveying & Deformation Disaster Monitoring of Hunan Province, Changsha, Hunan, China
[3]Key Laboratory of Metallogenic Prediction of Nonferrous Metals and Geological Environment Monitoring Ministry of Education, School of Geoscience and Info-physics, Central South University, Changsha, Hunan, China
[4]Department of Land Surveying and Geo-Informatics, Hong Kong Polytechnic University, Hong Kong, China
[5]Hunan Province Mapping and Science and Technology Investigation Institute, Changsha, Hunan, China

*Correspondence to*: Biyan Chen (yeary124@csu.edu.cn)

**Abstract.** Surface pressure ($P_s$) and weighted mean temperature ($T_m$) are two necessary variables for the accurate retrieval of precipitable water vapor (PWV) from global navigation satellite system (GNSS) zenith total delay (ZTD) estimates. The lack of $P_s$ or $T_m$ information is a concern for those GNSS sites that are not collocated with meteorological sensors. This paper investigates an alternative method of inferring accurate $P_s$ and $T_m$ at the GNSS station using nearby synoptic observations. $P_s$ and $T_m$ obtained at the nearby synoptic sites are interpolated onto the location of GNSS station by performing both vertical and horizontal adjustments, in which the parameters involved in $P_s$ and $T_m$ calculation are estimated from ERA-Interim reanalysis profiles. In addition, we present a method of constructing high quality PWV maps through vertical reduction and horizontal interpolation of the retrieved GNSS PWVs. To evaluate the performances of the $P_s$ and $T_m$ retrieval, and the PWV map construction, GNSS data collected from 58 stations of the Hunan GNSS network and synoptic observations from 20 nearby sites in 2015 were processed to extract the PWV so as to subsequently generate the PWV maps. The retrieved $P_s$ and $T_m$, and constructed PWV maps were assessed by the results derived from radiosonde and the ERA-Interim reanalysis. The results show that (1) accuracies of $P_s$ and $T_m$ derived by synoptic interpolation are within the range of 1.7-3.0 hPa and 2.5-3.0 K, respectively, which are much better than the GPT2w model; (2) the constructed PWV maps have good agreements with radiosonde and ERA-Interim reanalysis data with the overall accuracy being better than 3 mm; and (3) PWV maps can well reveal the moisture advection, transportation and convergence during heavy rainfall.

## 1 Introduction

Water vapor is an important meteorological parameter, which plays a crucial role in the formation of various weather phenomenon such as cloud, rain and snow (Ahrens and Samson, 2011). Water vapor accounts for only 0.1-3% of the total atmosphere mass, however due to the latent heat release, a small amount of water vapor may cause severe weather changes

(Mohanakumar, 2008). The monitoring of atmospheric water vapor variation is thus of significant value for short-term severe weather forecasting (Brenot et al., 2013; Labbouz et al., 2013; Van Baelen et al., 2011; Zhang et al., 2015). Among the various atmosphere sensing techniques, Global Navigation Satellite System (GNSS) is regarded as a uniquely powerful means to estimate the water vapor with advantages of all-weather capability, high accuracy and low-operating expenses (Bevis et al.,

1992; Guerova et al., 2016; Yao et al., 2017).

While GNSS signals are transmitted from satellites to ground receivers, they are delayed by the terrestrial troposphere. In GNSS data processing, the tropospheric delay is usually expressed as the zenith tropospheric delay (ZTD) multiplied by a mapping function, and sometimes plus horizontal gradients for a better GNSS positioning performance (Lu et al., 2016). The accuracy of the GNSS ZTD estimates depends on the data processing strategies and on the global products used in the

processing. At present, ZTDs are likely to be determined with accuracies up to several millimeters by a wide range of GNSS processing software (Pacione et al., 2017; Yuan et al., 2014). ZTD is normally divided into two parts: the zenith hydrostatic delay (ZHD) which is caused by the dry gases of the troposphere and the zenith wet delay (ZWD) which stems from the water vapor. The ZHD can be accurately calculated using empirical models with surface pressure ($P_s$) measured by meteorological sensors (Saastamoinen, 1972). ZWD is readily obtained with the subtraction of ZHD from ZTD. The precipitable water vapor

(PWV) can then be retrieved from ZWD with a conversion factor which is a function of the weighted mean temperature ($T_m$). $T_m$ can be calculated by numerical integration from the vertical profiles of atmospheric temperature and humidity (Davis et al., 1985). PWV is a key parameter in studying water vapor variations during severe weather phenomena, since it can reflect the inflow and outflow of water vapor in a vertical air column above a certain area (Yao et al., 2017).

As stated above, the retrieval of PWV from GNSS-ZTD needs two key meteorological parameters: $P_s$ and $T_m$. The first choice

is to measure the $P_s$ by a barometer collocated at the GNSS station. However, a large number of GNSS stations have been deployed for positioning purposes and are not equipped with collocated meteorological sensors. In this case, one may use pressure derived from a global atmospheric reanalysis (Dee et al., 2011; Zhang et al., 2017) or interpolated from nearby meteorological observations (Alshawaf et al., 2015; Musa et al., 2011; Wang et al., 2007) or predicted by a blind model (Böhm et al., 2015; Wang et al., 2017). For $T_m$, since the temperature and humidity profiles are very difficult to obtain, particularly in

a near-real-time mode, $T_m$ has to be calculated from a model. An empirical $T_m$ model dependent on surface temperature ($T_s$) (Bevis et al., 1994; Li et al., 2018) or a blind model developed from atmospheric reanalysis products (Böhm et al., 2015; Yao et al., 2013; Zhang et al., 2017) are often employed.

The work presented in this paper is carried out for constructing high quality PWV maps by a regional GNSS network in the Hunan Province, China for precipitation forecasts and analysis. The constructed high quality PWV maps will also be of

significant value for monitoring and early warning of geological disasters, such as landslides and mud-rock flows. In such a near-real-time application, the use of reanalysis products is not feasible. $P_s$ and $T_m$ have to be determined only using a blind model or nearby surface synoptic stations. The use of blind models is a very convenient means, however, most blind models (e.g. Global Pressure and Temperature 2 wet, GPT2w; (Böhm et al., 2015)) are developed at a global scale and are not likely to capture regional small-scale variations. More accurate $P_s$ and $T_m$ could be achieved by interpolation from nearby

meteorological observations if they can be accessed simultaneously. In this study, we investigate the construction of PWV maps from GNSS observations over the Hunan Province by performing the following five tasks: (1) $T_m - T_s$ relationship and vertical reduction models for $P_s$ and $T_m$ are developed for each synoptic station; (2) $P_s$ and $T_m$ data interpolated by nearby meteorological observations are compared with those derived from radiosonde and GPT2w models; (3) PWV vertical reduction model is developed for each GNSS station; (4) PWV interpolation is performed over the whole Hunan region and evaluated by radiosonde and European Centre for Medium Range Weather Forecasts Reanalysis (ECMWF) ERA-Interim reanalysis (hereafter short as ERA-I); and (5) the water vapor variation during a heavy rain event that occurred over a wide range of Hunan is examined based on PWV maps.

This paper is organized as follows. Section 2 presents the study area and the datasets used in the study. Section 3 describes the methodology to retrieve PWV from GNSS data. The strategy for meteorological data interpolation, $T_m$ modeling and PWV interpolation is also presented in this section. The assessment of $P_s$ and $T_m$ interpolated by nearby synoptic observations is described in section 4. The PWV maps constructed by GNSS data and PWV evolution during a heavy rain event are also presented in section 4. The summary and conclusions are given in section 5.

## 2 Study area and data description

The Hunan Province is located in the middle reaches of the Yangtze watershed in South Central China, with a territory of about 211,800 km$^2$. Hunan enjoys a subtropical humid monsoon climate bearing obvious continental climate features . The average annual rainfall varies between 1200-1700 mm, with 50%-60% concentrating in the months from April to August. Heavy showers and thunderstorms frequently occur in summer, causing catastrophic conditions as well as significant damages to urban infrastructure and agricultural production. The monitoring of water vapor variations using the GNSS network has a great potential to improve the capacity of extreme weather forecasting in the Hunan region.

### 2.1 GNSS, synoptic and radiosonde stations in Hunan

In 2015, 58 GNSS stations were deployed in the Hunan GNSS network and new stations have subsequently been added (see Figure 1). At present, the GNSS network consists of more than 90 stations and the number is still increasing (Li et al., 2017). Most of the GNSS stations are equipped with Trimble or Leica receivers and have a typical sampling interval of 30 s. In this study, the ZTDs are estimated using GNSS precise point positioning (PPP) technique with the Bernese 5.2 software (Dach et al., 2015). To examine their performance in real-time applications, IGS (International GNSS Service) ultra-rapid satellite orbit data and clock corrections are adopted in PPP processing. The ZTDs are estimated with an interval of 30 min, whilst the horizontal gradients are estimated every 12 h. The global mapping function (GMF) is used (Boehm et al., 2006) in the estimation, and GNSS observations with elevation angles below 5$^\circ$ are rejected. Evaluation results show that our estimated ZTDs have an accuracy of ~9 mm in the comparison by radiosonde measured ones. However, some stations in the Hunan GNSS network are not collocated with meteorological sensors, thus they cannot be directly used for water vapor monitoring.

Except the Hunan GNSS network, there are many GNSS stations without meteorological observations distributed across the province, which could be included for enhancing the quality of constructed PWV maps in the future. Therefore, a strategy of using nearby synoptic observations is needed to acquire the necessary meteorological parameters for GNSS-PWV retrieval. As shown in Figure 1, a total of 20 synoptic sites situated in Hunan and surrounding provinces can be used for this study. The average distance between a synoptic station and a GNSS station is about 41 km. The 6-hourly pressure and temperature data measured at the synoptic sites can be retrieved from the National Center for Atmospheric Research (NCAR) (http://rda.ucar.edu/datasets/ds336.0/). For real-time applications, pressure and temperature data at a given epoch are extrapolated from empirical models established using the past 20-day data. Here, the 4-order Fourier function is adopted for the empirical models. In addition, quality-assured atmospheric profiles observed by three radiosonde sites (marked with black diamonds in Figure 1) from the Integrated Global Radiosonde Archive (IGRA) (Durre et al., 2006) will be used to evaluate the meteorological data and PWV measurements. Both RSCS and RSCZ stations are equipped with GTS1 radiosonde sensors, whilst the type of GZZ2 sensor is adopted by the RSHH station.

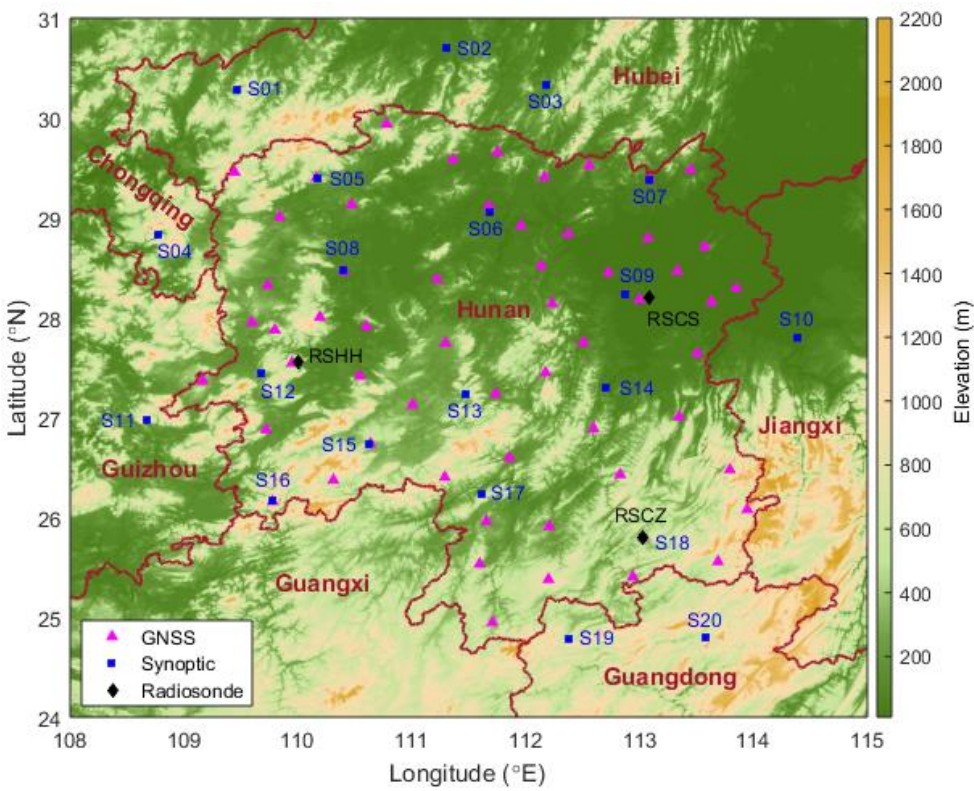

**Figure 1: Geographic distribution of GNSS, synoptic and radiosonde stations in Hunan and surrounding provinces.**

## 2.2 ECMWF reanalysis

ECMWF ERA-Interim is a global atmospheric reanalysis from 1979, continuously updated in near real time. In the reanalysis data generation, meteorological observations from in situ platforms (e.g., surface weather stations, ships, buoys, radiosonde stations and aircraft) and remote sensing satellites are assimilated into atmospheric physical models to recreate the past atmospheric conditions (Dee et al., 2011). Due to its high quality and global coverage, the ERA-Interim reanalysis has been exploited in various fields, e.g. GNSS meteorology (Wang et al., 2017; Zhang et al., 2017) and climate change research (Chen and Liu, 2016b; Lu et al., 2015). The ERA-I provides pressure, temperature, humidity and many other meteorological variables at 37 isobaric levels from 1000 hPa to 1 hPa with a 6 h interval. The reanalysis contains grid products with 11 different scales from 0.125°×0.125° to 3°×3°, and the horizontal resolution of 0.25°×0.25° is selected for this study, which equals to about 26 km in Hunan.

## 2.3 GPT2w model

The global pressure and temperature (GPT) model, which is developed using spherical harmonics (Boehm et al., 2007), can provide pressure and temperature at any site in the vicinity of the Earth's surface. Lagler et al., (2013) significantly improved the GPT model, especially for its spatial and temporal variability, and named this new version as GPT2. An extension version called GPT2w was developed by Böhm et al., (2015) with improved capability to determine ZWD in blind mode. Besides the pressure and temperature, the refined GPT2w model also provides various parameters such as water vapor pressure, weighted mean temperature and the temperature lapse rate.

## 3 PWV map construction with GNSS network observations

### 3.1 Retrieval of PWV from GNSS-ZTD

To retrieve PWV from GNSS inferred ZTD, ZHD should be determined first. The ZHD calculation formula is theoretically derived based on the assumption that the air is an ideal gas and that the troposphere satisfies the hydrostatic equilibrium (Davis et al., 1985). Saastamoinen (1972) derived the most widely used ZHD model as follows (Chen and Liu, 2016a):

$$\text{ZHD} = 2.2793\, P_s / (1 - 0.0026 \cos 2\varphi - 0.00028h), \tag{1}$$

where $\varphi$ is the station latitude (unit: radians) and $h$ is the height of the station above sea level (unit: km). By subtracting ZHD from ZTD, the remainder ZWD can then be converted to PWV by using the formula below (Askne and Nordius, 1987):

$$\text{PWV} = \frac{10^5}{(k_3/T_m + k_2')R_v} \text{ZWD}, \tag{2}$$

where $k_3 = 3.776 \times 10^5$ K$^2$/hPa, $k_2' = 16.52$ K/hPa and $R_v = 461.495$ J/K/kg are physical constants (Rüeger, 2002). The weighted mean temperature $T_m$ is defined as (Davis et al., 1985):

$$T_m = \frac{\int \frac{e(h)}{T(h)} \mathrm{d}h}{\int \frac{e(h)}{T(h)^2} \mathrm{d}h}, \tag{3}$$

where $e(h)$ and $T(h)$ are the water vapor pressure (hPa) and temperature (K) at height $h$, respectively. Since the humidity and temperature profiles are usually unavailable, a linear relationship between surface temperature $T_s$ and $T_m$ is often adopted to determine the $T_m$ :

$$T_m = a + bT_s, \tag{4}$$

where $a$ and $b$ are coefficients that need to be fitted locally using radiosonde or reanalysis profiles.

### 3.2 Spatial adjustments for $P_s$ and $T_m$

Because some stations in the Hunan GNSS network are not equipped with meteorological sensors, a method of spatially adjusting nearby meteorological observations to the GNSS stations was developed. Adjacent synoptic sites within the 100 km

radius of a given GNSS station are employed in the adjustments. The adopted radius ensures at least one synoptic site being located within the circumference centred on the GNSS station. For each GNSS site, on average, two synoptic stations fall into that circumference. First, surface pressure and mean weighted temperature data at the synoptic sites are adjusted to the height $H_s$ of the given GNSS station (Zhang et al., 2017):

$$P_s = P_r e^{\mu(H_s - H_r)}, \tag{5}$$

$$T_m = T_{mr} + \alpha(H_s - H_r), \tag{6}$$

where $P_r$, $T_{mr}$, and $H_r$ are the pressure (hPa), weighted mean temperature (K), and height (km) at the synoptic site, respectively. Here, $T_{mr}$ is calculated by equation (4) using the surface temperature (K) measured on site. $P_s$ and $T_m$ are the pressure and weighted mean temperature corresponding to the height $H_s$ at the synoptic site. $\mu$ and $\alpha$ are parameters needed to be estimated at the synoptic site.

Then the vertically adjusted meteorological data are interpolated to the location of the GNSS station according to:

$$y_G = \frac{\sum_{i=1}^{n} exp(-d_i^2) \cdot y_i}{\sum_{i=1}^{n} exp(-d_i^2)}, \tag{7}$$

where $n$ is the number of synoptic sites with a distance less than 100 km to the given GNSS site; $y_G$ is the interpolated value; $y_i$ is the adjusted meteorological data at synoptic site $i$; and $d_i$ is the distance between synoptic site $i$ and the GNSS station.

### 3.3 PWV interpolation from GNSS stations

With the use of interpolated $P_s$ and $T_m$, PWV data at the GNSS stations could be obtained in near-real-time. In order to construct the PWV map, GNSS PWV data are used to interpolate at a $0.25° \times 0.25°$ grid. Similar to the meteorological data, PWVs at nearby GNSS stations are interpolated to the given height $H_p$ of the grid point as follows (Dousa and Elias, 2014):

$$\text{PWV} = \text{PWV}_r \left[ 1 - \frac{\beta(H_s - H_r)}{T_s} \right]^{\frac{\theta \cdot g}{\beta \cdot R_d}}, \tag{8}$$

where $\text{PWV}_r$ is the PWV estimated at the GNSS station; $\beta$ refers to the temperature lapse rate (unit: K/km); $\theta$ a numerical coefficient; $g$ is gravity acceleration (unit: $m \cdot s^{-2}$); and $R_d$=287.053 $J \cdot K^{-1} \cdot kg^{-1}$ is the gas constant for dry air. Both $\beta$ and $\theta$ are required to be determined from local observations for a better performance. The PWV at the grid point can then be acquired by interpolation using equation (7). In this study, the height of each grid point is derived from the global topography/bathymetry grid that has a 30-arc second resolution (SRTM30 PLUS) (Becker et al., 2009).

## 4 Results and discussion

### 4.1 Evaluation of $P_s$ and $T_m$ interpolated by synoptic data

All the parameters including $a$ and $b$ in equation (4), $\mu$ in (5), and $\alpha$ in (6) are estimated locally at each synoptic site using reanalysis products. In this study, the values of $a$, $b$, $\mu$ and $\alpha$ (their values are given in Table 1) for each site are fitted from ERA-I atmospheric profiles over the whole year of 2014. With the use of the estimated parameters, spatial adjustments for $P_s$ and $T_m$ to radiosonde stations are performed throughout the year of 2015. Then the interpolated meteorological data are directly compared with the radiosonde observations.

Table 1 Estimated values of $a$, $b$, $\mu$ and $\alpha$ for the 20 synoptic sites using ERA-I atmospheric profiles over the whole year of 2014

| Station | Parameters | | | |
|---------|--------|------|---------|-------|
|         | $a$ | $b$ | $\mu$ | $\alpha$ |
| S01 | 264.72 | 0.82 | -0.1110 | -4.47 |
| S02 | 264.40 | 0.83 | -0.1112 | -4.48 |
| S03 | 264.90 | 0.82 | -0.1106 | -4.25 |
| S04 | 267.08 | 0.75 | -0.1102 | -3.76 |
| S05 | 265.67 | 0.79 | -0.1111 | -4.05 |
| S06 | 266.46 | 0.78 | -0.1104 | -3.90 |
| S07 | 265.68 | 0.79 | -0.1103 | -4.16 |
| S08 | 266.49 | 0.77 | -0.1108 | -3.79 |
| S09 | 267.32 | 0.73 | -0.1101 | -3.88 |
| S10 | 267.23 | 0.73 | -0.1102 | -4.09 |
| S11 | 269.07 | 0.67 | -0.1097 | -3.66 |
| S12 | 267.99 | 0.72 | -0.1105 | -3.66 |

| | | | | |
|---|---|---|---|---|
| S13 | 268.40 | 0.70 | -0.1105 | -3.64 |
| S14 | 268.74 | 0.65 | -0.1074 | -4.06 |
| S15 | 269.02 | 0.68 | -0.1103 | -3.69 |
| S16 | 269.56 | 0.66 | -0.1099 | -3.78 |
| S17 | 269.43 | 0.66 | -0.1102 | -3.70 |
| S18 | 269.52 | 0.65 | -0.1099 | -3.96 |
| S19 | 270.27 | 0.63 | -0.1096 | -4.04 |
| S20 | 269.82 | 0.64 | -0.1094 | -4.14 |

Figure 2 shows the time series of $P_s$ provided by radiosonde, synoptic interpolation and GPT2w model at three radiosonde stations over 2015. Surface pressures interpolated from synoptic observations have a very good agreement with radiosonde measured ones. The GPT2w model basically reflects the variation trend of $P_s$ throughout the year, however, it is unable to capture the fluctuations which are especially obvious in winter and spring months. Similar results can be observed in Figure 3 for $T_m$ comparison. Detailed statistics of the comparison results are given in Table 2. RMS (root mean squares) errors of $P_s$ and $T_m$ derived from synoptic interpolation vary in the range of 1.7-3.0 hPa and 2.5-3.0 K, respectively. In comparison, the RMS errors from the GPT2w model are 4.7-5.6 hPa and 3.8-4.2 K, respectively, for $P_s$ and $T_m$, which are much larger than the synoptic interpolation method. In terms of maximum and minimum differences, GPT2w derived values are significantly larger than those derived from synoptic interpolation, further indicating the GPT2w model is less accurate. According to equation (1), 1 hPa error in surface pressure would cause about 2.3 mm error in ZHD. Therefore, 3 hPa error in $P_s$ will result in an error of about 6.9 mm in ZHD (~1.15 mm in PWV). The relative error of the PWV caused by the $T_m$ error is approximately equal to the relative error of the $T_m$ (Zhang et al., 2017). Derived from Figure 3 and Table 2, the relative error of synoptic data interpolated $T_m$ is about 1%. In the study region, the PWV value is usually less than 80 mm, meaning the PWV error caused by $T_m$ error is within 0.8 mm. In addition, as mentioned in section 2.1, our estimated ZTDs have an accuracy of about 9 mm (~1.45 mm in PWV). On the whole, the accuracy of PWV retrieved from GNSS-ZTD using $P_s$ and $T_m$ from synoptic interpolation is better than 3.4 mm. Following the same error analysis, the uncertainty of PWV caused by GPT2w model is about 4.6 mm.

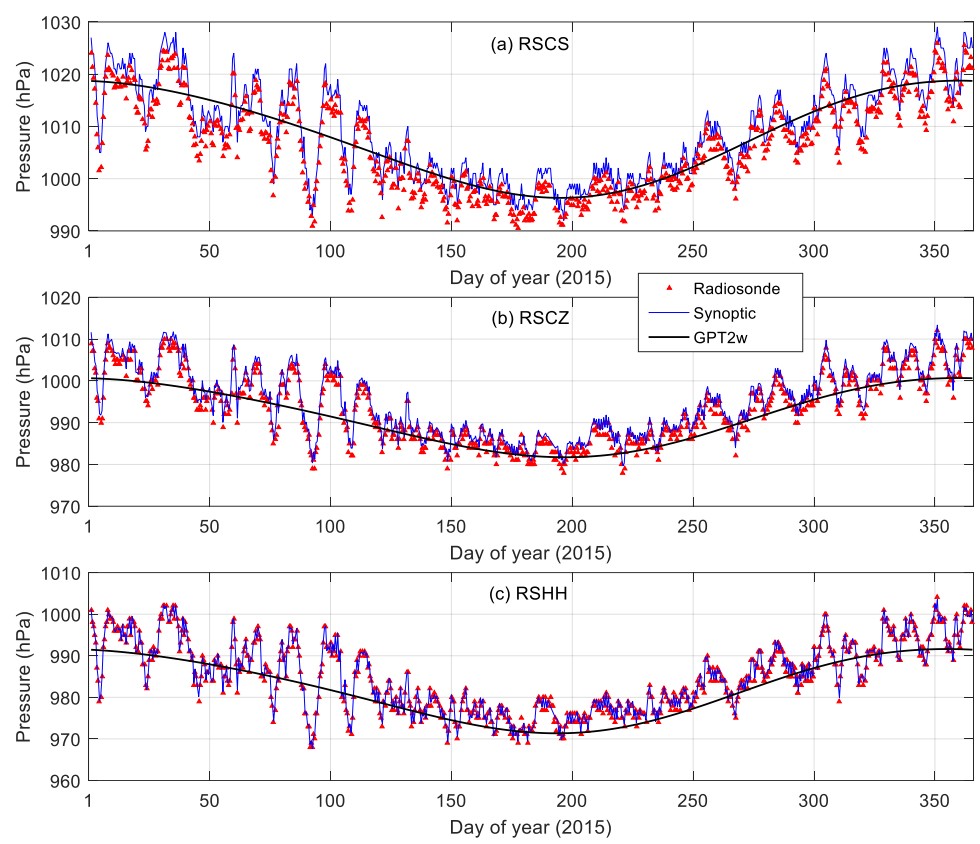

**Figure 2:Time series of surface pressure provided by radiosonde, synoptic adjustment and GPT2w model over the whole year of 2015 at three radiosonde stations: (a) RSCS, (b) RSCZ, and (c) RSHH, all of which are located in the Hunan Province, China.**

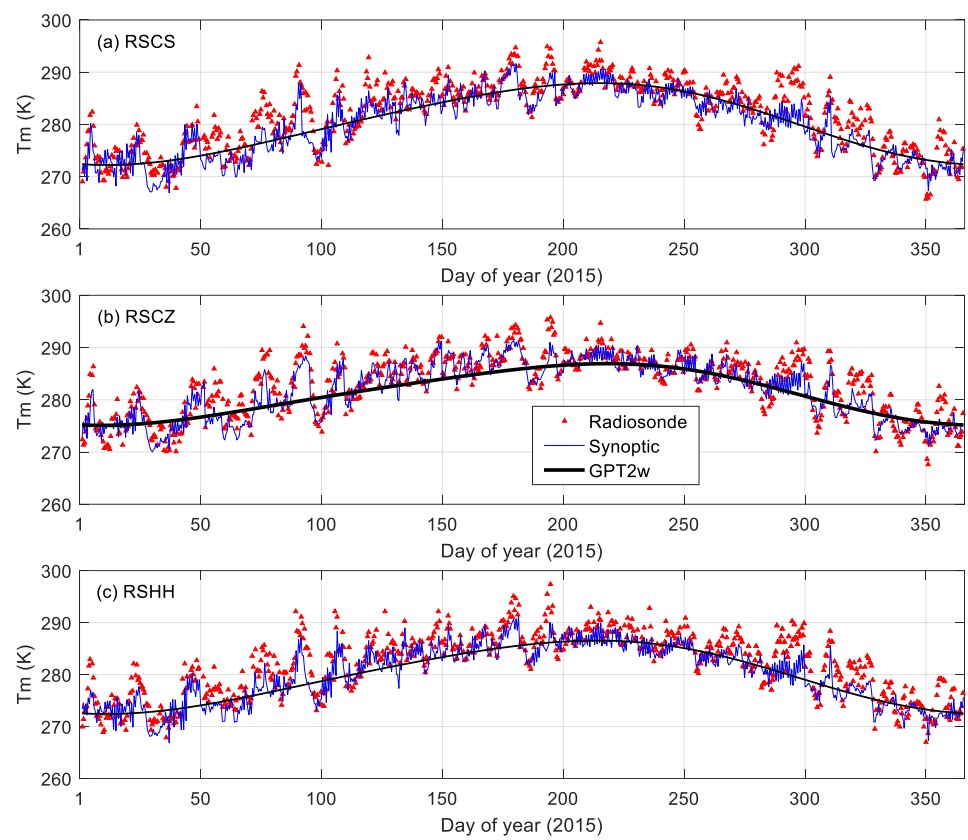

**Figure 3: Time series of weighted mean temperature provided by radiosonde, synoptic adjustment and GPT2w model over the whole year of 2015 at three radiosonde stations: (a) RSCS, (b) RSCZ, and (c) RSHH, all of which are located in the Hunan Province, China.**

**Table 2: Comparison of $P_s$ and $T_m$ for Radiosonde-Synoptic and Radiosonde-GPT2w at the three radiosonde stations**

| Comparison | | RSCS | | RSCZ | | RSHH | |
|---|---|---|---|---|---|---|---|
| | | Ps (hPa) | Tm (K) | Ps (hPa) | Tm (K) | Ps (hPa) | Tm (K) |
| Radiosonde *vs* Synoptic | Bias | 2.91 | 1.47 | -1.66 | 1.14 | -2.58 | 1.49 |
| | RMS | 2.97 | 2.92 | 1.74 | 2.58 | 2.61 | 2.76 |
| | Max | 5.04 | 9.69 | 0.48 | 7.42 | -1.33 | 9.17 |
| | Min | 1.01 | -6.40 | -3.82 | -5.18 | -3.82 | -5.40 |
| Radiosonde *vs* GPT2w | Bias | 1.23 | 1.59 | 2.02 | 1.68 | 3.06 | 2.23 |
| | RMS | 4.70 | 3.84 | 4.76 | 4.02 | 5.56 | 4.16 |
| | Max | 13.75 | 13.29 | 12.26 | 14.21 | 14.96 | 14.48 |
| | Min | -16.13 | -7.44 | -13.46 | -8.08 | -14.94 | -6.34 |

**4.2 Evaluation of PWV by radiosonde**

At each GNSS station, GNSS-derived ZTDs are converted to PWVs with the use of meteorological parameters interpolated from synoptic data. In order to evaluate the GNSS-derived PWV, the GNSS PWVs are interpolated onto the radiosonde stations according to equations (7) and (8) for a direct comparison with radiosonde measured ones. As displayed in Figure 4, GNSS interpolated PWVs agree well with the radiosonde measured ones at all the three radiosonde stations. Mean biases of the PWV differences at RSCS, RSCZ and RSHH station are -0.59 mm, 1.04 mm and 1.40 mm, respectively (see Table 3). In terms of the RMS error, they are 2.47 mm, 2.94 mm and 2.69 mm for RSCS, RSCZ and RSHH stations, respectively. It is notable that for weather nowcasting, the accuracy threshold is 3 mm (Yuan et al., 2014). This indicates that the GNSS-derived PWVs are accurate enough for the application of weather nowcasting. Additionally, the probability density function (PDF) of PWV differences and the fractional error as percent by radiosonde 5 mm PWV bins are exhibited in Figure 5. As shown in Figure 5(a), about 83% PWV differences are within the range of -5~5 mm. The fractional errors vary from about -15% to 6% as radiosonde PWV increases from 0 mm to 75 mm. When PWV values are less than 10 mm, there is an obvious wet bias relative to the radiosonde. This is probably related to the dry bias of Chinese made GTS1 and GZZ2 radiosonde sensors caused by solar heating (Moradi et al., 2013). Whereas, an obvious dry bias can be observed for PWV values larger than 65 mm. The dry bias is likely due to the overestimation of water vapor by radiosonde as the humidity sensors suffer contamination from rain and clouds during radiosonde ascents (Bock et al., 2005; Nash et al., 2011).

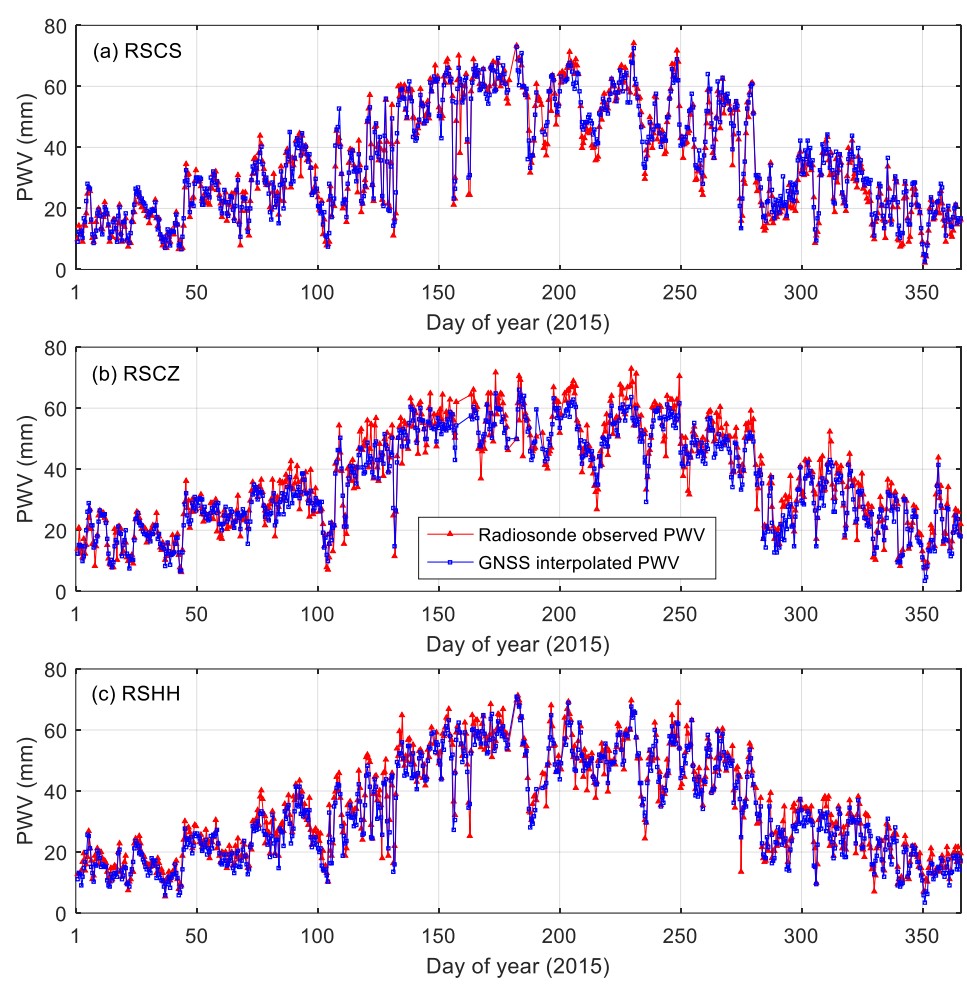

**Figure 4: Time series of PWV measured by radiosonde and interpolated by GNSS over the whole year of 2015 at three radiosonde stations: (a) RSCS, (b) RSCZ, and (c) RSHH.**

**Table 3: Comparison between radiosonde observed and GNSS interpolated PWV at the three radiosonde stations**

| Radiosonde station | Bias (mm) | RMS (mm) | Max (mm) | Min (mm) |
|---|---|---|---|---|
| RSCS | -0.59 | 2.47 | 7.46 | -7.96 |
| RSCZ | 1.04 | 2.94 | 10.44 | -11.15 |
| RSHH | 1.40 | 2.69 | 9.27 | -8.59 |

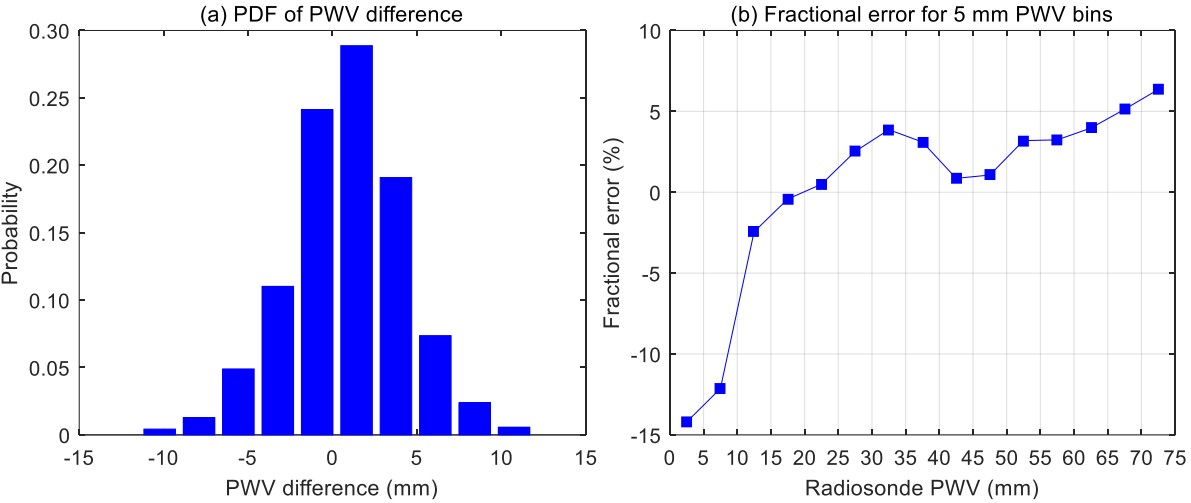

**Figure 5: (a) PDF of PWV difference and (b) fractional error as percent by radiosonde 5 mm PWV bins. All the three radiosonde stations are used in the statistics.**

### 4.3 PWV comparison between ERA-I products and GNSS interpolation data

The ERA-I products are used to further assess the performance of PWV maps constructed by the GNSS network data. For the comparison, the GNSS PWVs are interpolated onto grid points with a spatial resolution of 0.25° in both latitude and longitude directions to match the ERA-I PWV data. Figure 6 presents the spatial distribution of the bias and RMS error of the PWV differences between ERA-I and GNSS over the Hunan region. As seen in Figure 6(a), the bias varies from -8 mm to 6 mm depending upon the location. In general, mountainous regions have a larger bias than plain regions. In terms of RMS error, as

shown in Figure 6(b), its values vary in the range of 2–8 mm. Large parts of the studied region are populated with RMS errors less than 3 mm. However, relatively large RMS errors of more than 6 mm are obtained for some mountainous regions.

In addition, the PDF of the PWV differences shown in Figure 7(a) indicate that there is a higher probability of negative PWV difference. Negative values account for about 64% of the total PWV difference. The fractional error as percent by ERA-I 5 mm PWV bins varies greatly from about -65% to 10%. When PWV values are smaller than 10 mm, there is an obvious wet

bias relative to the ERA-I. The largest negative fractional error occurs at the extremely low (less than 5 mm) PWV values. When PWV values are larger than 60 mm, dry bias relative to ERA-I can be observed for PWV values. Figure 7(c) exhibits the relationship between RMS error and elevation. It is clearly seen that the RMS error increases generally with increase in elevation. A high correlation coefficient of 0.73 is achieved between RMS error and elevation. This is consistent with the bias and RMS error maps in Figure 6. The high correlation coefficient is probably due to reasons: 1) vertical adjustment for PWV

according to equation (8) is unable to accurately capture the highly dynamic water vapor variation in the vertical direction; and (2) the performance of the ERA-I PWV product degrades in mountainous regions due to the larger errors caused by PWV averaging over cells with highly variable surface topography (Alshawaf et al., 2017).

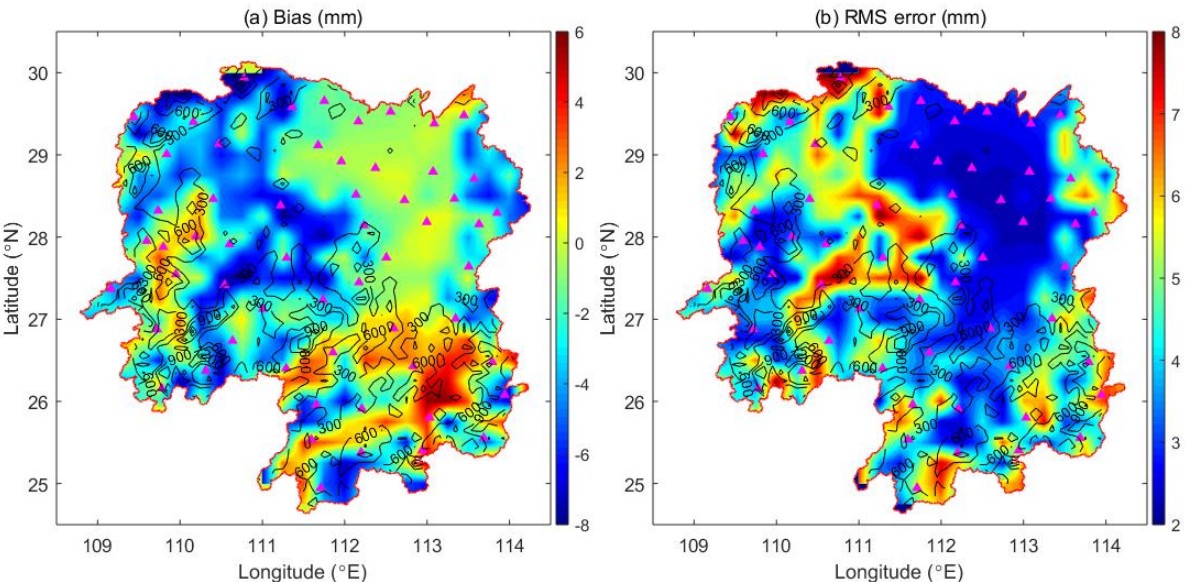

**Figure 6: Map of (a) bias and (b) RMS error of the differences between ERA-I PWV and GNSS interpolated PWV over the Hunan Province for the year 2015. Black contours represent the elevation (unit: m).**

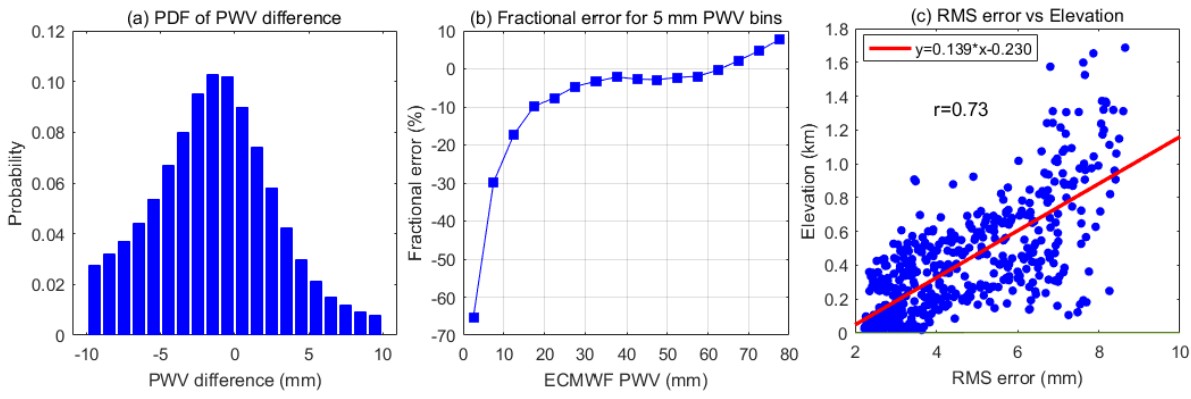

**Figure 7: (a) PDF of PWV difference, (b) fractional error as percent by ERA-I 5 mm PWV bins, and (c) relationship between RMS error and elevation for the comparison between ERA-I and GNSS.**

### 4.4 Monitoring water vapor variations using GNSS derived PWV maps

The ultimate goal of this study is to apply the constructed PWV maps for the study of weather forecasting. We further investigated the water vapor variations during a large-scale heavy precipitation event using the PWV maps derived from GNSS observations. In June 2015, Hunan Province suffered several large-scale regional torrential rains, which caused major floods and massive landslides in some places. An average rainfall of 236 mm over the whole province was recorded in that month, and the accumulated rainfall exceeded 500 mm in many areas. In this study, we focused on a heavy rainfall process occurring during 6-8 June 2015. Figure 8 exhibits the geographic distribution of the daily accumulated precipitation over the Hunan

Province for 6, 7 and 8 June 2015. The precipitation data are retrieved from the Tropical Rainfall Measuring Mission (TRMM), a joint mission of NASA (National Aeronautics and Space Administration) and the Japan Aerospace Exploration Agency to measure rainfall for weather and climate research (Kummerow et al., 1998; Lau and Wu, 2011). As shown in Figure 8(a), the accumulated precipitation on 6 June decreased from about 60 mm at the southeast to 0 mm at the northwest. On 7 June, rainfalls were observed over most parts of the province with heavy precipitation mainly occurring in the northern Hunan Province. Afterwards, on 8 June, the precipitation weakened on most of the Hunan province except for an increase in the northeast.

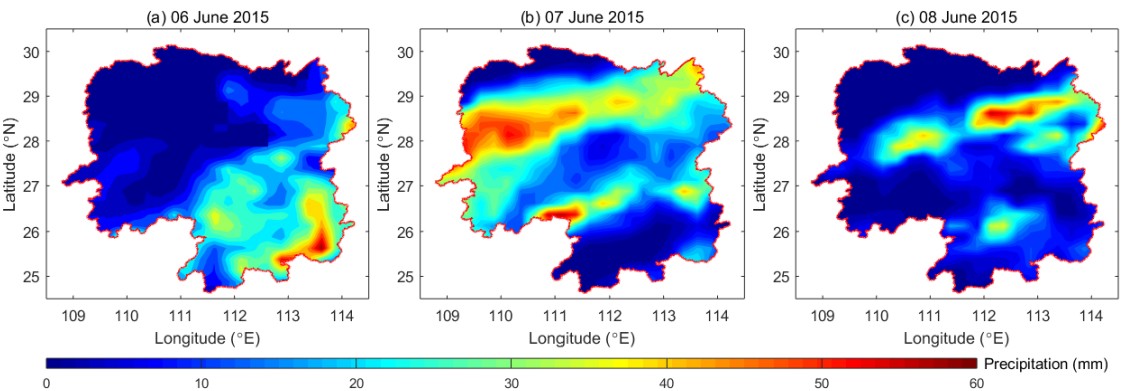

**Figure 8: Map of daily accumulated precipitation in the Hunan Province on (a) 6 June 2015, (b) 7 June 2015, and (c) 8 June 2015. The precipitation data were retrieved from the TRMM with a spatial resolution of 0.25°×0.25°.**

Figure 9 presents the evolution of PWV derived from GNSS observations for the Hunan Province during the period of 6-8 June 2015 with a time interval of 6 h. In addition, the TRMM derived rain rates over Hunan for the same epochs are displayed in Figure 10. On 6 June (see Figure 9(a-d)), the whole province experienced an obvious increase in PWV from south to north, indicating that a large amount of moisture from the south flowed into Hunan. This is consistent with the precipitation pattern displayed in Figure 8(a) in that the rainfall gradually decreased from south to north. On 7 June, significant PWV changes mainly concentrated in regions north of 28°N. Especially in the northeast, PWV experienced an increase of 10-15 mm from UTC 00 to UTC 12 of 7 June and then dissipated quickly. On 8 June, obvious PWV decreases were observed in the northwest, whereas the southeast experienced a slight increase in PWV. The precipitation maps shown in Figure 8(b) and (c) also agree well with the PWV variations. From 7 June to 8 June, the precipitation areas largely decreased in the north whilst slightly expanded in the south. Referring to the rain rates at the corresponding epochs, as shown in Figure 10, we cannot observe close correlations between the PWV and the rain rate. Larger moisture convergence is not necessarily linked with higher rain rate occurrence. This is because the moisture convergence is not the only cause of precipitation, whilst also controlled by many other factors such as wind, temperature and terrain. However, the GNSS-derived PWV maps are able to reveal the moisture advection, transportation and convergence during the heavy precipitation event.

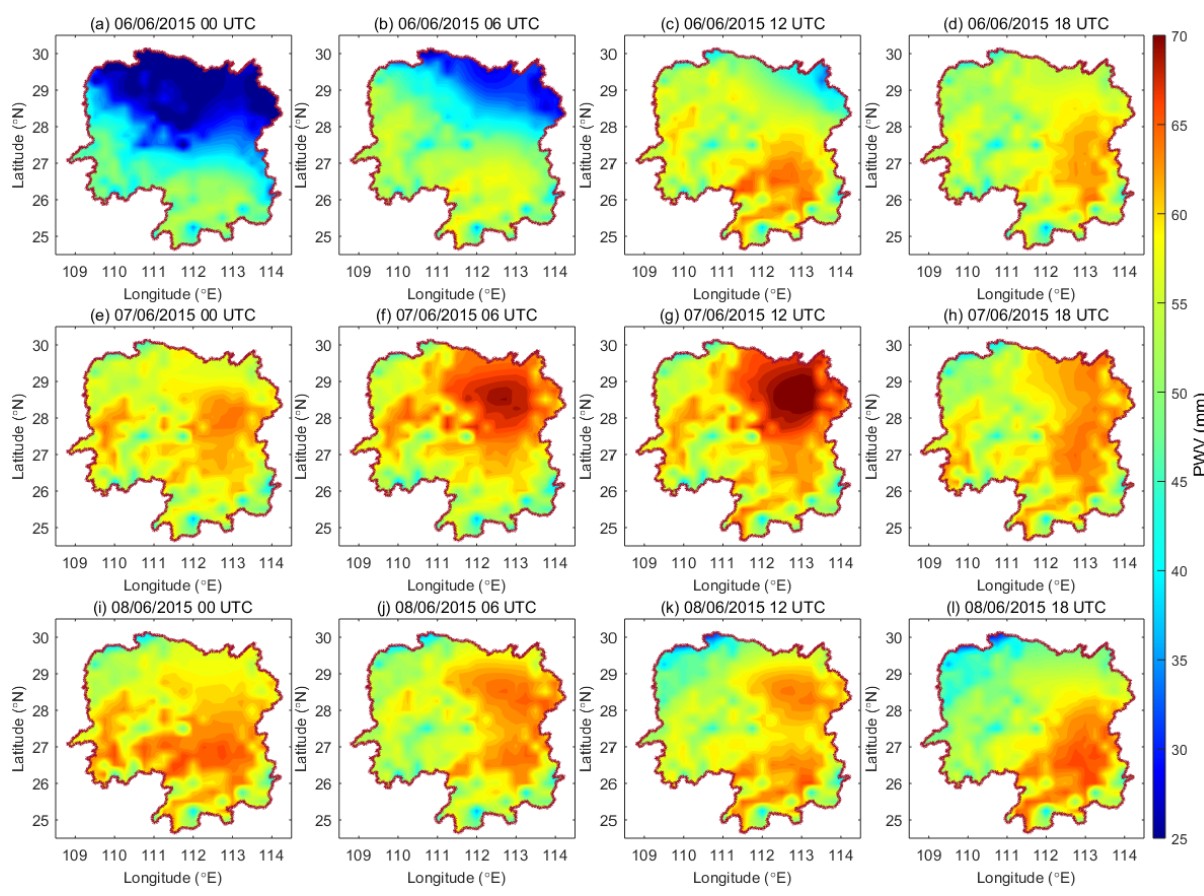

**Figure 9: Evolution of GNSS-derived PWV maps for the Hunan province every 6 h from UTC 00, 6 June 2015 to UTC 18, 8 June 2015.**

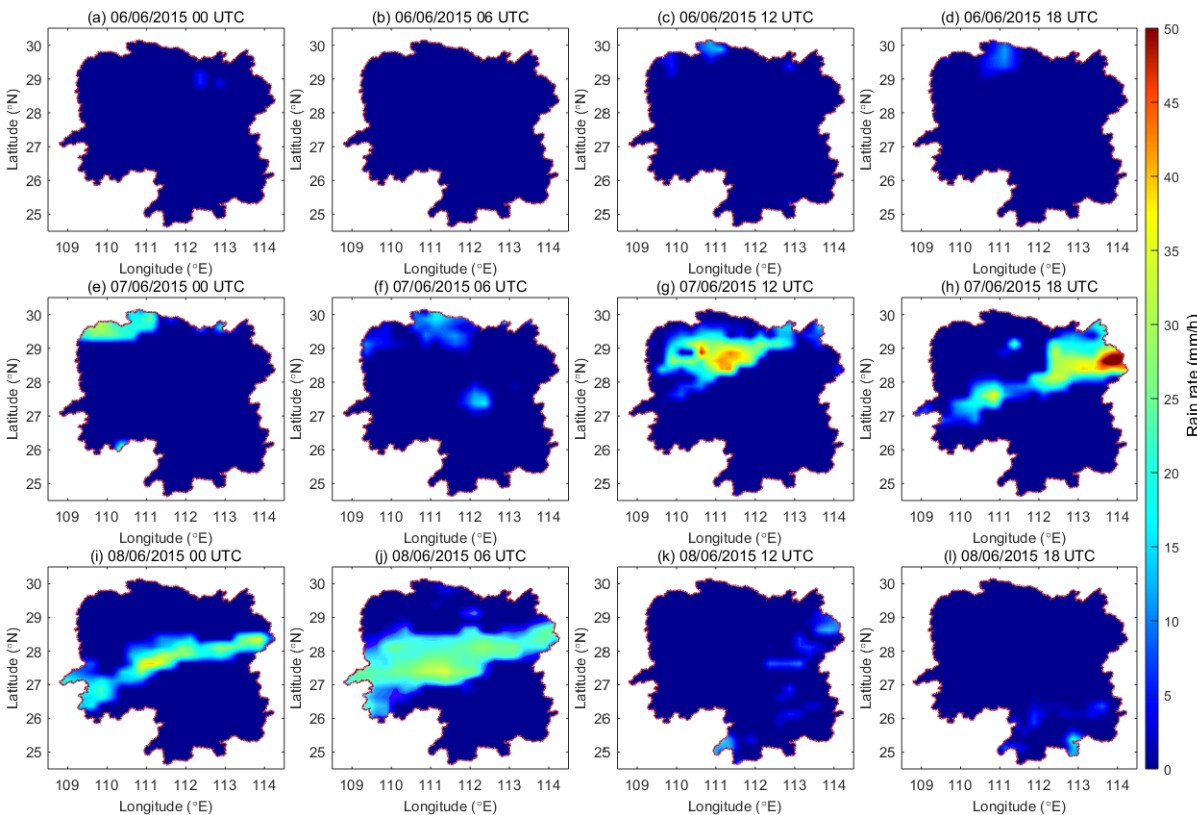

**Figure 10: Evolution of rain rate maps for the Hunan province every 6 h from UTC 00, 6 June 2015 to UTC 18, 8 June 2015. The rain rate data were retrieved from the TRMM with a spatial resolution of 0.25°×0.25°.**

5 In addition, Figure 11 further exhibits the geographic distribution of the correlation coefficient between precipitation and PWV. The correlation coefficients vary greatly from -0.9 to 0.8 depending upon the location. High positive correlation coefficients are present in western regions between 27°N and 27.5°N. Precipitation and PWV show a high negative relationship in eastern regions between 26°N and 27°N. It can be observed from Figure 1 and Figure 11 that high positive/negative correlation coefficients mainly occur in mountainous regions, especially in hillsides and valleys. This is because the meso-scale orography

10 creates favorable conditions for precipitation formation by generating moisture convergence and the small scale orography plays an important role by triggering convective initiation and enhancement (Labbouz et al., 2013). Therefore, precipitation and PWV correlate more closely in mountainous regions than flat terrains, and mountainous regions are often sensitive areas prone to high frequency of heavy precipitation.

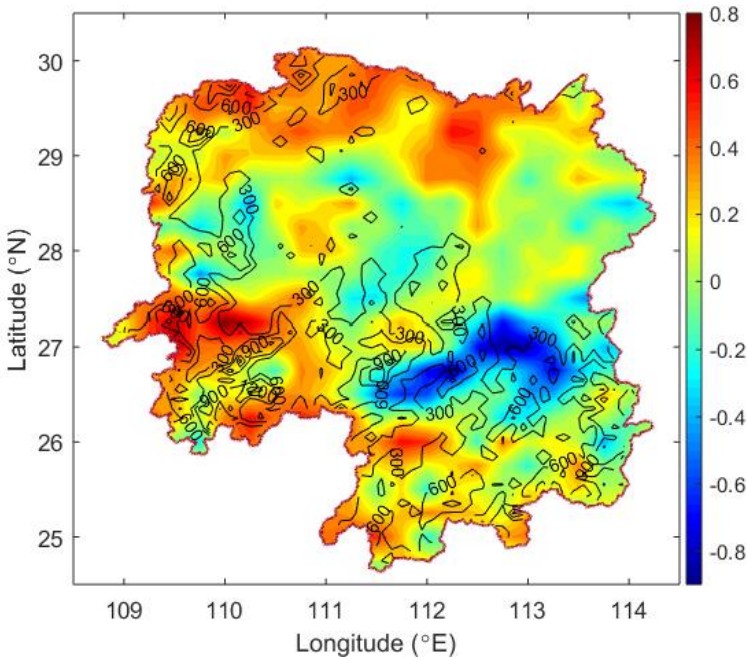

**Figure 11: Map of the correlation coefficient between precipitation and PWV for the heavy rainfall process of 6-8 June 2015 over the Hunan Province. Black contours represent the elevation (unit: m). The correlation coefficients were calculated for the accumulated precipitation and PWV for the 6-8 June 2015 period.**

## 5 Summary and conclusions

The lack of collocated meteorological data at GNSS station makes it difficult to take full advantage of GNSS observations for weather studies. This paper investigates an alternative method for accurate determination of PWV for near-real-time applications using GNSS data and nearby synoptic observations. Moreover, we present a method to construct PWV maps with the use of a GNSS network, which is critical for improving the forecasting capability of extreme weathers, e.g. heavy rainfall. The proposed approach for PWV map construction consists of two main steps: 1) the $P_s$ and $T_m$ derived at the nearby synoptic sites are interpolated onto the location of the GNSS stations through both vertical and horizontal adjustments; and 2) vertical reduction and horizontal interpolation are performed to construct PWV map using the retrieved GNSS PWV. In this study, ERA-I data over the whole year of 2014 were employed to estimate all the parameters involved in the above two steps. The accuracies of the synoptic interpolated and GPT2w derived $P_s$ and $T_m$ have been evaluated by comparing them against the observed values at 3 radiosonde sites in 2015. RMS errors of $P_s$ and $T_m$ derived from the GPT2w model vary in the range of 4.7-5.6 hPa and 3.8-4.2 K, respectively. The RMS errors from synoptic interpolation are 1.7-3.0 hPa and 2.5-3.0 K, respectively, which are much better than the GPT2w model.

In addition, GNSS interpolated PWVs are assessed with respect to reference PWV values from radiosonde and ERA-I. GNSS interpolated PWVs show a good agreement with the radiosonde measured ones with RMS errors varying in the range of 2.4-

3.0 mm. In the comparison with ERA-I, the biases of their differences vary from -8 mm to 6 mm over the Hunan Province and mountainous regions have a larger bias than flat regions in general. RMS errors are within the range of 2–8 mm with those for most regions being less than 3 mm. For PWV values less than 10 mm or more than 60 mm, there is an obvious wet or dry bias relative to ERA-I. Furthermore, the RMS errors are found to increase with increased elevation in general and a high correlation

coefficient of 0.73 is obtained between RMS error and elevation.

We further apply the constructed PWV maps to monitor the water vapor variability during a large-scale heavy precipitation event that occurred during 6-8 June 2015 in the Hunan Province. Results demonstrate that it is possible to reveal the moisture advection, transportation and convergence during the heavy rainfall using PWV maps. Since the orography provides favorable conditions for precipitation formation, we also find that the precipitation and PWV correlate more closely in mountainous

regions, especially in hillsides and valleys.

This research demonstrates the potentials of retrieving accurate PWV from GNSS observations using adjacent synoptic data and generating high-quality PWV maps from the GNSS network for weather prediction in near-real-time. Future work will focus on the three following issues: (1) examining the reliability of the PWV map construction in other areas with highly dynamic water vapor; (2) assessing the performance of the constructed PWV maps with higher spatial and temporal resolutions;

and (3) assimilating the PWV maps into a numerical prediction model to enhance the capability of extreme weather forecasting.

*Data availability*. The ECMWF ERA-Interim reanalysis products are available online (http://apps.ecmwf.int/datasets/). The radiosonde data were obtained from http://weather.uwyo.edu/upperair/sounding.html. The TRMM rainfall data were provided by https://pmm.nasa.gov/data-access/downloads/trmm. The synoptic observations were provided by

http://rda.ucar.edu/datasets/ds336.0/. The SRTM30 PLUS data were provided by http://topex.ucsd.edu/index.html. The radiosonde data of Hong Kong were obtained from http://weather.uwyo.edu/upperair/sounding.html. The GNSS observations of the Hunan GNSS network presented in this study are available from the authors upon request (yeary124@csu.edu.cn).

*Competing interests*. The authors declare that they have no conflict of interest.


*Acknowledgments*. This work was supported by the Research Grant for Specially Hired Associate Professor of Central South University (project No.: 202045005). Zhizhao Liu thanks the Hong Kong Polytechnic University (projects 152149/16E, 152103/14E, 152168/15E, and 1-BBYH) and the grant supports from the Key Program of the National Natural Science Foundation of China (project No.: 41730109). The European Centre for Medium-Range Weather Forecasts is appreciated for

providing the ECMWF reanalysis data. The TRMM rainfall data were provided by the National Aeronautics and Space Administration (NASA), via https://pmm.nasa.gov/data-access/downloads/trmm. The synoptic observations were provided by the National Center for Atmospheric Research (NCAR), from the website http://rda.ucar.edu/datasets/ds336.0/. The SRTM30 PLUS data were provided by the Satellite Geodesy research group at the Cecil H. and Ida M. Institute of Geophysics and Planetary Physics, Scripps Institution of Oceanography, University of California San Diego, from the website

http://topex.ucsd.edu/index.html. Finally, the authors want to thank the University of Wyoming for providing the radiosonde data.

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
