# Peer review of "Constructing Precipitable Water Vapor Map from Regional GNSS Network Observations without Collocated Meteorological Data for Weather Forecasting"

_Atmospheric Measurement Techniques, 2018_

## Referee Comment (RC1) · Anonymous Referee #1 · 2 Jun 2018

**Title:** Constructing Precipitable Water Vapor Map from Regional GNSS Network Observations without Collocated Meteorological Data

**Authors:** Biyan Chen, Wujiao Dai, Zhizhao Liu, Lixin Wu, Cuilin Kuang, and Minsi Ao

Ps and Tm are required during the conversion process from zenith wet delay (ZWD) to precipitable water vapour (PWV) from GNSS measurements. An alternative method for accurate determination of PWV was proposed for near-real-time applications using GNSS data and nearby synoptic observations, and a method to construct PWV maps with the use of GNSS network was presented. The demand for assimilating zenith total delay (ZTD) into numerical weather prediction (NWP) models is very high in meteorological practice, in particular its role in weather now-casting. The research conducted is of importance to improve weather predictions over the region investigated in HuNan China.

Overall I have no serious concerns with respect to the manuscript. Your analysis is sound and you show plenty of results upon which you draw your conclusions. Also the English is very good and generally your statements are clear. However, there are a few minor points I would like to raise (in no particular order):

1. Page 3 LINE 2, You stated that "None of them is however collocated with meteorological sensors"? Actually, more than 70 stations have meteorological observations collected by HuNan Meteorological Bureau. You need either remove this statement or make a different statement. In fact the HuNan CORS network does have significant amount of collocated radiosonde measurements since 2015. They are typically easily accessible.
2. In Figure 1, it seems RSCZ radiosonde station collocated with one of the CORS stations. Pls compare the CORS derived PWV with RS-derived PWV.
3. In page4, line 14, "The blind model global pressure and temperature (GPT)", this sentence should be changed in a better order.
4. In page5, line 25, where do the formula (5-6) come from? Please show the references and the unit of parameters.
5. In section 4.1 please show the formula (4-8) coefficients estimated locally at each synoptic site using reanalysis products.
6. All the PWV maps show minor color variations, which is difficult to show the large moisture variations during a rainstorm. I would recommend add 2 more colors between blue and yellow, such as red and purple.
7. In Figure 9, the PWV maps should be compared to precipitation every 6 hours. The temporal span of 24h in Figure 10 is too large to explain the situation effectively.
8. Two recent journal publications to do the same region (and perhaps similar data and data sources are used) in China should be consulted. Their publication details are listed below.
   - Li L, Wu S, Wang X, Tian Y, He C, and Zhang K (2018) Modelling of weighted-mean temperature using regional radiosonde observations in Hunan China, Terr. Atmos. Ocean. Sci., Vol.29,No.2,187-199, doi: 10.3319/TAO.2017.05.26.01.
   - Li LI, Suqin Wu, Xiaoming Wang, Ying Tian, Changyong He and Kefei Zhang (2017) Seasonal Multi-Factor Modelling of Weighted-Mean Temperature for Ground-Based GNSS Meteorology in Hunan, China, Advances in Meteorology, volume 17, https://doi.org/10.1155/2017/3782687.

---

## Referee Comment (RC2) · Anonymous Referee #2 · 14 Jun 2018

Dear Authors, This is a very interesting manuscript which applies well-used methods to the GNSS-derived ZTD from Hunan, China. I have no major comments on the scientific contents. It is well written with minor grammatical and spelling mistakes. However, they are a few of which some I have pointed out in the annotated manuscript I have uploaded (amt-2018-083-supplement.pdf).

General comments:

1) You do not mention how the GNSS data have been processed or where the solution is from. Clearly GNSS-derived ZTD are not raw observations and a couple of sentences on this step or a reference pointing to details of the GNSS processing strategy

are required.

2) Figure 1. Change the colour scheme of this figure to something more commonly used for the presentation of topography. For example, low lying areas should be in green and high areas in brown (green-orange-yellow-red-brown). Also, I did not notice the legend at first. It might be useful for other readers if you box it in and make the background of the legend white.

3) You have only employed the GPT2 model. It is well known that this model can only reflect the annual variation in p and t and not the daily fluctuations. For this the values from the VMF1 model (although derived from ECMWF) would be more adequate for the comparison. As there is a VMF1 model for forecasts, you could also employ this in near-real-time.

4) Section 4.2. You mention the wet/dry biases between the PWVs from GNSS and radiosonde data. There are references out there and you should mention that other authors have found similar biases, linking your work more to previously published work. Where do the biases come from?

5) Additional reference, place as indicated. Guerova, G., J. Jones, J. Dousa, G. Dick, S. de Haan, E. Pottiaux, O. Bock, R. Pacione, G. Elgered, H. Vedel and M. Bender (2016). "Review of the state-of-the-art and future prospects of the ground-based GNSS meteorology in Europe." Atmos. Meas. Tech., 9, 5385-5406, https://doi.org/10.5194/amt-9-5385-2016, 2016.

6) minor comment: often when you talk about ECMWF you actually mean the ERA-Interim reanalysis, hence ERA-I would be a better abbreviation.

Please also note the supplement to this comment:
https://www.atmos-meas-tech-discuss.net/amt-2018-83/amt-2018-83-RC2-supplement.pdf

**Supplement:**

[revised manuscript text omitted]

---

## Referee Comment (RC3) · Anonymous Referee #3 · 30 Jun 2018

General Comments.

The manuscript describes a method to convert GNSS-derived Zenith Total Delay into Precipitable Water Vapour using, for each GNSS stations, surface pressure and weighted mean temperature of the atmosphere obtained interpolating nearby synoptic observations. The analysis is well presented and the results sound reasonable. However, I would raise the following issues which has to be clarified prior to the publication.

1. In the paper, there is no indication on how GNSS data are analyzed: which strategy is applied for estimating ZTD? Which global products are used? What is the ZTD sampling rate? What is the accuracy of the GNSS ZTD estimates? What is the latency

of GNSS ZTD estimates?

2. The authors claim they are presenting a method inferring 'accurate' Ps and Tm and for the construction of 'high-quality' PWV maps. Both 'accurate' and 'high-quality' has to be quantified with respect to the target application the authors are interested in. This because the observational requirements are different according to the different target application. I would suggest reviewing the title by adding in it the target application of this research.

Below specific comments.

Line 13 pag.1. My suggestion is to replace '(GNSS) data' with '(GNSS) Zenith Total Delay (ZTD) estimates'

Line 25 pag.1 replace 'ERA reanalysis' with 'ERA-Interim reanalysis'

Line 4 pag.2 Suggested reference: Guerova, G., Jones, J., Douša, J., Dick, G., de Haan, S., Pottiaux, E., Bock, O., Pacione, R., Elgered, G., Vedel, H., and Bender, M.: Review of the state of the art and future prospects of the ground647 based GNSS meteorology in Europe, Atmos. Meas. Tech., 9, 5385-5406, doi:10.5194/amt-9-5385-2016, 2016.

Line 7 pag.2 'for a better performance', please clarify this statement.

Line 8 pag.2 See general comment. The accuracy of the GNSS ZTD estimates depend on how the data are processed and on the global products used in the processing. For example, the agreement of reprocessed ZTD estimates is at 2 mm level (reference Pacione, R., Araszkiewicz, A., Brockmann, E., and Dousa, J.: EPN Repro2: A reference GNSS tropospheric dataset over Europe, Atmos. Meas. Tech., 10, 1689–1705, doi: 10.5194/amt-2016-369, 2017).

Line 24 pag.3 what is the average distance between a synoptic station and a GNSS station?

Line 26 pag.3 I guess the ZTD sampling rate is higher than 6h, right? If so, how do you interpolate in time pressure and temperature data measured at the synoptic station? What is the error of this interpolation? Such error has to be added in the error analysis done in section 4.1.

Line 28 pag.3 What kind of radiosonde are used?

Line 12 pag.5 Different set of refractivity coefficients are available in literature, please add the reference about the used ones.

Line 18 pag.5 The empirical model of eq.4 suffers from diurnal and seasonal biases, are such biases acceptable for the considered application?

Line 22 pag. 5 Could the authors explain on which ground they chose 100 km as the radius of the circumference centred on the GNSS site? On average, how many synoptic stations fall into that area for each GNSS sites?

Line 7 pag.6. Considering eq.7, what is the interpolation error?

Line 10 pag. 7 Why in this error analysis the authors are not considering the ZTD error? The ZTD error is of course the same in both models the authors are evaluating but I think has to be considered in the total error budget.

Line 1 pag. 10 Replace 'measured' with 'estimated'

In the manuscript several times, ECMWF should be replaced with ERA-Interim. The quality of the maps should be improved. Fig. 8a check the white spot
* * *

---

## Author Comment (AC1) · 12 Jul 2018

Dear Reviewer

Thank you very much for your time and effort towards our manuscript. We have revised the manuscript greatly according to your invaluable suggestions and comments. Please find our detailed responses and revisions in the supplement files.

Best regards

The Authors

[Figure]

Please also note the supplement to this comment:
https://www.atmos-meas-tech-discuss.net/amt-2018-83/amt-2018-83-AC1-supplement.zip
* * *

---

## Author Comment (AC2) · 12 Jul 2018

Dear Reviewer

Thank you very much for your time and effort towards our manuscript. We have revised the manuscript greatly according to your invaluable suggestions and comments. Please find our detailed responses and revisions in the supplement files.

Best regards The Authors

Please also note the supplement to this comment:

[Figure]

https://www.atmos-meas-tech-discuss.net/amt-2018-83/amt-2018-83-AC2-supplement.zip

---

## Author Response (AR1)

Dear Editors,

Thanks very much for your valuable time and clear instructions on our manuscript. We have revised our manuscript carefully based on the valuable comments and suggestions from the three reviewers. Please kindly find our point-to-point responses below.

Best regards,

The authors

**Anonymous Referee #1**

Received and published: 2 June 2018

see my comments in the attached PDF file

Please also note the supplement to this comment:

https://www.atmos-meas-tech-discuss.net/amt-2018-83/amt-2018-83-RC1-supplement .pdf

Ps and Tm are required during the conversion process from zenith wet delay (ZWD) to precipitable water vapour (PWV) from GNSS measurements. An alternative method for accurate determination of PWV was proposed for near-real-time applications using GNSS data and nearby synoptic observations, and a method to construct PWV maps with the use of GNSS network was presented. The demand for assimilating zenith total delay (ZTD) into numerical weather prediction (NWP) models is very high in meteorological practice, in particular its role in weather now-casting. The research conducted is of importance to improve weather predictions over the region investigated in HuNan China. Overall I have no serious concerns with respect to the manuscript. Your analysis is sound and you show plenty of results upon which you draw your conclusions. Also the English is very good and generally your statements are clear. However, there are a few minor points I would like to raise (in no particular order):

R. Thanks for your positive conclusions towards our manuscript.

1. Page 3 LINE 2, You stated that "None of them is however collocated with meteorological sensors"? Actually, more than 70 stations have meteorological observations collected by HuNan Meteorological Bureau. You need either remove this statement or make a different statement. In fact the HuNan CORS network does have significant amount of collocated radiosonde measurements since 2015. They are typically easily accessible.

R. Thanks for your useful information. At present, we are not able get the meteorological observations collected by Hunan Meteorological Bureau. In the future, we will try to collaborate with the Hunan Meteorological Bureau to get those data. We thus revised the sentence 'None of them is however collocated with meteorological sensors' to 'However, some stations in the Hunan GNSS network are not collocated with meteorological sensors, thus they cannot be directly used for water vapor monitoring. Except for the Hunan GNSS network, there are many GNSS stations without meteorological observations distributed across the province, which could be included for enhancing the quality of constructed PWV maps in the future'.

2. In Figure 1, it seems RSCZ radiosonde station collocated with one of the CORS stations. Pls compare the CORS derived PWV with RS-derived PWV.

R. Thanks for this comment. Yes, the distance between the RSCZ radiosonde station and the GNSS station is only 266 m, and their height difference is 8 m. In section 4.2, the RSCZ radiosonde-derived PWVs were compared with those interpolated from surrounding 4 GNSS stations including the collocated station using the following equation:

$$y_G = \frac{\sum_{i=1}^n exp(-d_i^2) \cdot y_i}{\sum_{i=1}^n exp(-d_i^2)}$$

We can get that the larger the distance, the smaller the weight. Except the collocated GNSS stations, the distances of the other 3 stations to the radiosonde stations all exceed 50 km. Therefore, the weight for the collocated GNSS station is greater than 0.99 in the PWV calculation. For this reason, the Figure 4(b) can be seen the direct PWV comparison between radiosonde and corresponding GNSS station. The RMS error of their PWV differences is 2.94 mm. As you suggested, we also compare the radiosonde PWVs with the collocated GNSS PWVs, the statistical results are basically the same.

Thank you very much again.

3. In page4, line 14, "The blind model global pressure and temperature (GPT)", this sentence should be changed in a better order.

R. Thanks for your correction. We have revised the sentence 'The blind model global pressure and temperature (GPT)' to 'The global pressure and temperature (GPT) model'.

4. In page5, line 25, where do the formula (5-6) come from? Please show the references and the unit of parameters.

R. These two formulas are from the following journal paper:

Zhang, H., Yuan, Y., Li, W., Ou, J., Li, Y. and Zhang, B.: GPS PPP-derived

precipitable water vapor retrieval based on Tm/Ps from multiple sources of meteorological data sets in China, J. Geophys. Res. Atmospheres, doi:10.1002/2016JD026000, 2017.

Note that the original formula for the  $P_s$  in the reference is as follows:

$$P_{s} = P_{r} e^{\frac{-gM(H_{s}-H_{r})}{RT_{v}}}$$

We slightly revise the above formula to  $P_s = P_r e^{\mu(H_s - H_r)}$  to refine the modeling of the  $P_s$ . In addition, we cited the above paper when describing the formula (5-6) and gave the unit of parameters.

Thank you very much.

5. In section 4.1 please show the formula (4-8) coefficients estimated locally at each synoptic site using reanalysis products.

R. Thanks for your comment. The estimated parameters using the ERA-I reanalysis products over the year of 2014 are shown in the following table. We have added this table in section 4.1.

| Station.    |        | Para | meters  |       |
|-------------|--------|------|---------|-------|
| Station     | а      | b    | μ       | α     |
| S 01 | 264.72 | 0.82 | -0.1110 | -4.47 |
| S02  | 264.40 | 0.83 | -0.1112 | -4.48 |
| S03  | 264.90 | 0.82 | -0.1106 | -4.25 |
| S04         | 267.08 | 0.75 | -0.1102 | -3.76 |
| S05         | 265.67 | 0.79 | -0.1111 | -4.05 |
| S 06 | 266.46 | 0.78 | -0.1104 | -3.90 |
| S07  | 265.68 | 0.79 | -0.1103 | -4.16 |
| S08  | 266.49 | 0.77 | -0.1108 | -3.79 |
| S09  | 267.32 | 0.73 | -0.1101 | -3.88 |
| S 10 | 267.23 | 0.73 | -0.1102 | -4.09 |
| S 11 | 269.07 | 0.67 | -0.1097 | -3.66 |
| S12         | 267.99 | 0.72 | -0.1105 | -3.66 |
| S13  | 268.40 | 0.70 | -0.1105 | -3.64 |
| S14         | 268.74 | 0.65 | -0.1074 | -4.06 |
| S15         | 269.02 | 0.68 | -0.1103 | -3.69 |
| S 16 | 269.56 | 0.66 | -0.1099 | -3.78 |
| S 17 | 269.43 | 0.66 | -0.1102 | -3.70 |
| S18  | 269.52 | 0.65 | -0.1099 | -3.96 |
| S19         | 270.27 | 0.63 | -0.1096 | -4.04 |

Table 1 Estimated values of a, b,  $\mu$  and  $\alpha$  for the 20 synoptic sites using ERA-I atmospheric profiles over the whole year of 2014

| S20 | 269.82 | 0.64 | -0.1094 | -4.14 |
|-----|--------|------|---------|-------|
|-----|--------|------|---------|-------|

6. All the PWV maps show minor color variations, which is difficult to show the large moisture variations during a rainstorm. I would recommend add 2 more colors between blue and yellow, such as red and purple.

R. Thanks very much for your useful comment. We changed the contour color for the PWV maps. Below shows the revised PWV maps. As you mentioned, it's more clear to show the moisture variations.

---

## Author Response (AR2)

Dear Dr. Roeland Van Malderen,

Thanks very much for your valuable time and clear instructions on our manuscript. We have revised our manuscript carefully based on the valuable comments and suggestions. Please kindly find our point-to-point responses below.

Best regards,

The authors

Associate Editor Decision: Publish subject to minor revisions (review by editor) (13 Aug 2018) by Roeland Van Malderen

Comments to the Author:

15 Dear Authors,

thank you very much for your dedication in answering almost all of the comments raised by the reviewers. Your paper has improved significantly because of this. I still have left some two minor, although important, issues to address:

20 * When you evaluate the GNSS-derived PWV with the PWV measured by radiosondes, you discuss the dry biases of the radiosondes, which is caused by solar heating (reference of Moradi et al. 2013). This dry bias is only try for certain types of radiosondes, so you should mention which type of radiosonde has been used at the three sites that you have used! Also the wet bias RS bias that you mention at high PWV ranges (reference of Bock et al. 2005) is radiosonde type dependent, see for instance a more receent publication

25 of the same author: https://www.atmos-meas-tech.net/6/2777/2013/. An alternative explanation (a

different sensitivity of GNSS and RS to different PWV ranges) of the GNSS-RS PWV differences is also given in e.g. https://www.atmos-meas-tech.net/7/2487/2014/. Of course, you can contact me for further details.

R: Thanks very much for your insightful comments. Both RSCS and RSCZ stations are equipped with GTS1 radiosonde sensors, whilst the type of GZZ2 sensor is adopted by the RSHH station. As reported by Moradi et al. (2013), the Chinese made radiosonde sensors also suffer dry bias caused by solar heating. As you suggested, we introduced the types of radiosonde sensors used at the three stations in both sections 2.1 and 4.2.

In addition, Nash et al. (2011) and many Chinese researchers reported that the Chinese made radiosondes can overestimate the humidity when the surface is very moist or passing through thick clouds. Thus, the dry bias of GNSS PWV is very likely due to the overestimation of water vapor by radiosonde as the humidity sensors suffer contamination from rain and clouds during radiosonde ascent.

*Nash, J., Oakley, T., Vömel, H. and Li, W.: WMO Intercomparison of high quality radiosonde systems, World Meteorological Organization: Instruments and observations, Yangjiang, China., 2011.*

* when you evaluate the GNSS-derived PWV with the ERA-I PWV, you make the rather general statement that the performance of the high-resolution (O.25°X0.25°) ERA-I PWV product degrades with increased elevation. Please give some references here, as the ERA-I PWV product has been already evaluated frequently and you should find some evidence for this statement in such previous studies. A recent study (in which other references are found) is e.g. https://www.atmos-chem-phys-discuss.net/acp-2018-137/

R: Thanks very much for your comments. We found a similar statement in a recent paper reported by Alshawaf et al., (2017) that the bias between the ERA-I and GNSS PWV data sets increases for sites in mountainous regions. We thus referred that paper in the revised manuscript and revised the sentence 'the performance of the high-resolution (0.25°×0.25°) ERA-I PWV product degrades with increased elevation'

to 'the performance of the ERA-I PWV product degrades in mountainous regions due to the larger errors caused by PWV averaging over cells with highly variable surface topography (Alshawaf et al., 2017)'.

*Alshawaf, F., Balidakis, K., Dick, G., Heise, S. and Wickert, J.: Estimating trends in atmospheric water vapor and temperature time series over Germany, Atmospheric Meas. Tech., 10(9), 3117–3132, doi:10.5194/amt-10-3117-2017, 2017.*

And then, some technical corrections:

* page 1, line 30: water vapor accounts for only 0.1-3% of the total atmosphere --> in mass? density? Please specify.

R: It is in mass. We have corrected it accordingly. Thanks

* page 2, line 4: to estimated instead of of to estimating

* page 2, line 4: low-operating expenses instead of low-operating expense?

* page 2, line 20: add a ":" after needs two key meteorological parameters

* page2, line 21: add an "a" before barometer

* page 2, line 22: add "are" before not equipped

* page 5, line 10: equals to instead of equates to

* page 6, line 10: please add "adopted radius" in between This ensures...

* page 15, line 22: Referring to instead of Refer to

R: Thanks very much. We have corrected them accordingly.

* page 17, line 8: I guess you want to add "eastern" to the regions where you find a high negative correlation between precipitation and PWV between 26°N and 27°N

R: Yes, we added an "eastern" before the 'regions' in the manuscript. Thanks.

5  * page 17, lines 8-9: please also refer here to your Figure 1, in which the elevation of the studied area is shown.

R: Thanks very much. As you suggested, we revised the sentence 'It can be found that high positive/negative correlation coefficients mainly occur in mountainous regions, especially in hillsides and valleys' to 'It can be observed from Figure 1 and Figure 11 that high positive/negative correlation
10  coefficients mainly occur in mountainous regions, especially in hillsides and valleys'.

* page 18, Figure 11: please mention in the figure caption if the correlation coefficients are calculated for the accumulated precipitation and accumulated PWV for the 6-8 June 2015 period.

R: Thanks very much. We have added the sentence 'The correlation coefficients were calculated for the
15  accumulated precipitation and PWV for the 6-8 June 2015 period' to the caption of Figure 11.

Thank you very much again. Your invaluable suggestions have further improved the quality of our manuscript.

[revised manuscript text omitted]

---

## Author Response (AR3)

Dear Dr. Roeland Van Malderen,

We have modified our manuscript based on your valuable suggestions. Thanks very much.

Best regards,

The authors

Associate Editor Decision: Publish subject to technical corrections (22 Aug 2018) by Roeland Van Malderen

Comments to the Author:

Dear authors,

just two small comments:

* on page 4, lines 11-12: please give a reference for the Chinese radiosonde types.

R: The radiosonde types used at the three stations can be found in the metadata of radiosonde provided by IGRA. We thus give the reference website there, i.e. https://www1.ncdc.noaa.gov/pub/data/igra/history/igra2-metadata.txt.

* on page 11, lines 15-16: please specify that the Chinese radiosonde types are known to have a moist bias for large PWV ranges (Nash et al. 2011 reference) and a similar effect has been found for other radiosonde types and was ascribed to the humidity sensors suffering from contamination from rain and clouds during radiosonde ascents (Bock et al. 2005 reference).

R: Thanks for your correction. We revised the sentence 'The dry bias is likely due to the overestimation of water vapor by radiosonde as the humidity sensors suffer from contamination from rain and clouds

during radiosonde ascents (Bock et al., 2005; Nash et al., 2011).' to 'The dry bias is likely due to the overestimation of water vapor by radiosonde as the Chinese radiosonde types are known to have a moist bias for large PWV ranges (Nash et al., 2011). A similar effect has also been found for other radiosonde types and was ascribed to the humidity sensors suffering from contamination from rain and clouds during radiosonde ascents (Bock et al., 2005).'.

And last, but not least: I would encourage you to write an e-mail to the AMT office and ask if the paper could be included in the GNSS special issue https://www.atmos-meas-tech.net/special_issue89.html. This would increase the visibility of your paper and research considerably, as all the recent papers dealing with GNSS tropospheric research can be found at one place, after one click!

R: Thanks for your suggestion. We have emailed the Editorial Support Svenja Lange. We are waiting for his/her response.

Thank you very much again for your comments.

[revised manuscript text omitted]